# *Coriocella* and the Worms: First Record of Scale-Worm *Asterophilia* cf. *culcitae* Ectosymbiotic on a Mollusc

**Giulia Fassio** [1,2]

1   Department of Biology and Biotechnologies "Charles Darwin", Sapienza University of Rome,
    00185 Rome, Italy; giulia.fassio@uniroma1.it
2   Institut de Systématique, Évolution, Biodiversité (ISYEB), UMR 7205 (CNRS, EPHE, MNHN, UPMC),
    Muséum National d'Histoire Naturelle, Sorbonne Universités, 75005 Paris, France

**Abstract:** Species of the mollusc genus *Coriocella* (Velutinidae) produce defensive biocompounds, making them potentially valuable hosts for other marine invertebrates. However, so far, only two instances of crustaceans ectosymbiotic on their mantle have been reported. This is the first observation, made in New Caledonia, of a pair of scale-worms identified as *Asterophilia* cf. *culcitae* (Polynoidae) hiding themselves on the mantle of *Coriocella* cf. *tongana*. This finding represents the first evidence of a symbiotic interaction between these two groups, expanding the association range for both taxa, and providing new insight into their, mostly unknown, ecology.

**Keywords:** symbiont; scale-worm; Velutinidae; Polynoidae; gastropod; New Caledonia; association; host range expansion

Lamellariinae d'Orbigny, 1841 (Mollusca: Velutinidae) is a subfamily of caenogastropods distributed worldwide, excluding the poles, from shallow to deep waters [1–4]. In all species within this subfamily, the shell has lost its defensive function, becoming a fragile and internal structure entirely enclosed by the mantle. This shell reduction is potentially linked to the abundance of pyrrole alkaloids in lamellariin mantles, which act as defensive biocompounds [5–7], making them good refugia for other marine invertebrates. However, to date, only two symbiotic associations have been reported for this subfamily, both involving crustaceans hosted by the snail genus *Coriocella* Blainville, 1824. This lamellariin genus is widely distributed in the Indo-West Pacific, easily recognizable by the presence of a variable number of dorsal mantle warts and a typical light brown or dark-velvet-like coloration. A no-better-identified amphipod was observed burrowing in the dorsal mantle of *Coriocella hibyae* Wellens, 1991, in Magoodhoo Island (Maldives) [8], and the shrimp *Zenopontonia rex* (Kemp, 1922) (Crustacea: Palaemonidae) was reported associated with *Coriocella nigra* Blainville, 1824, on South Button Island (India) [9].

The scale-worm family Polynoidae Kinberg, 1856 (Annelida) includes many species symbiotic with various marine invertebrates (e.g., cnidarians, decapods, sponges, echinoderms, and other polychaetes), with most specialized toward a single host, but some showing instead a broader range of hosts [10,11]. The polynoid genus *Asterophilia* Hanley, 1989 is characterized by dorsal cirri with white tips and red elytra with white or yellow spots, and it includes two species: *Asterophilia culcitae* Britayev & Fauchald, 2005, distributed from Southeast Asia to Japan, and *Asterophilia carlae* Hanley, 1989, present from Fiji to Southeast Asia [12]. These species are a well-known symbiont of asteroids, holothurians, and, more rarely, crinoids, and can be found living in pairs (female and male) on the same host, with females generally slightly larger and longer than males [12,13]. Body color can either enable or not camouflage on the host; when it does not, female bodies tend to be reddish, while male bodies are whitish [12–14].

Reported here for the first time are two specimens of the scale-worm *A.* cf. *culcitae* symbiotic with the lamellariin gastropod *Coriocella* cf. *tongana* (Quoy & Gaimard, 1832).

One specimen of lamellariin (MNHN-IM-2019-26118), with two scale-worms (MNHN-IA-2021-675 and MNHN-IA-2021-676) hidden in its mantle, was hand collected while snorkeling off Nouméa (New Caledonia; locality details in Supplementary Material Table S1) on 17/09/2022, during the scientific expedition "Quinzaine des Nudibranches 2022" organized by the Muséum national d'Histoire naturelle, Paris (MNHN). Both worms were hiding among the warts of the dorsal mantle of the snail (Figure 1a). One worm was smaller and whitish (MNHN-IA-2021-676, 5 mm in length; Figure 1c,d) and hidden in the left lateral side of the snail mantle, while the other was larger and reddish (MNHN-IA-2021-675, 10 mm in length; Figure 1b,e) and hidden in the posterior part of the mantle.

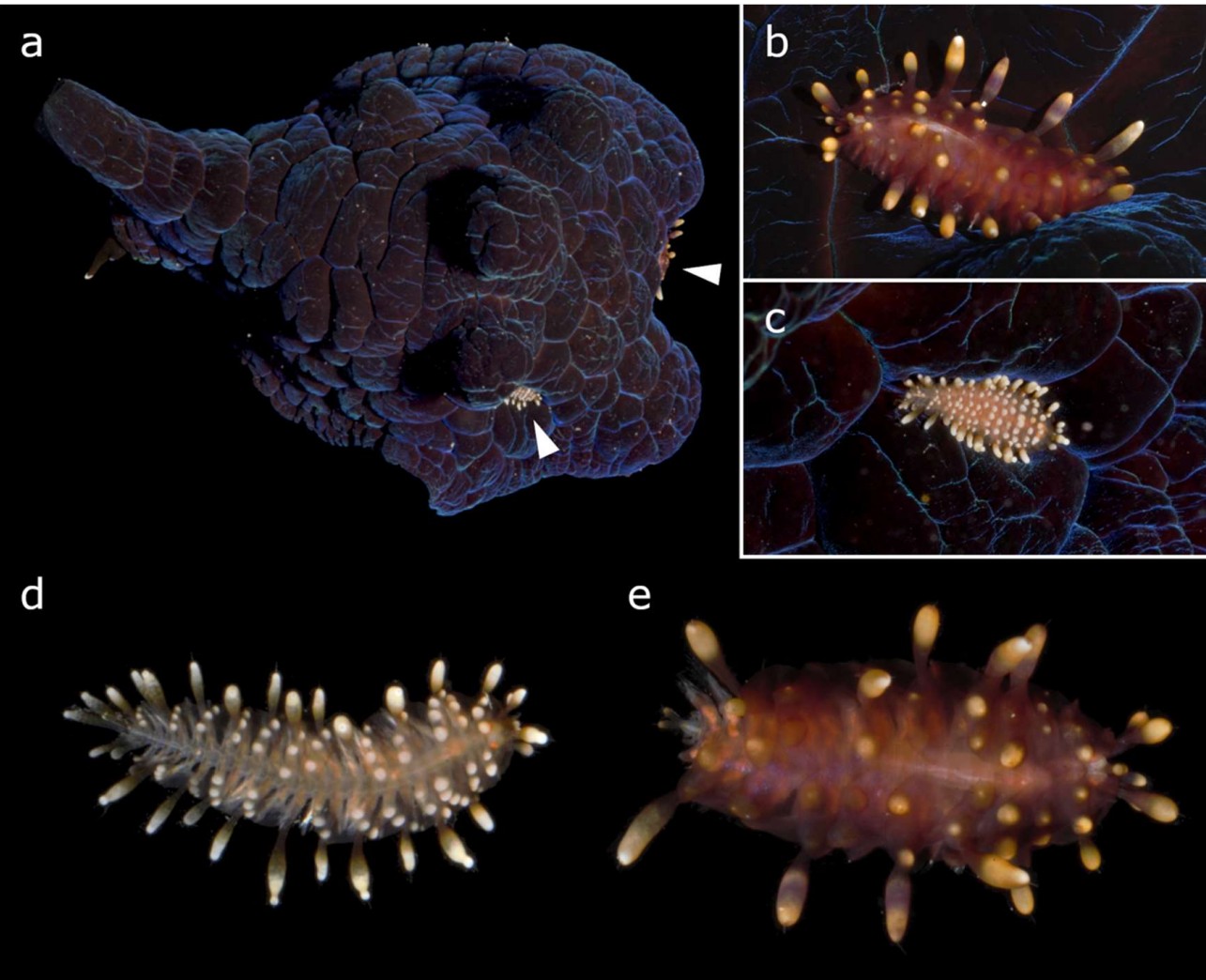

**Figure 1.** (**a**) *Coriocella* cf. *tongana* MNHN-IM-2019-26118 (50 mm in length) with two *Asterophilia* cf. *culcitae* hidden in the mantle (indicated by arrowheads). (**b**,**e**) *A.* cf. *culcitae* MNHN-IA-2021-676 (10 mm in length, 2 mm maximum width without parapodia, 6 mm with parapodia). (**c**,**d**) *A.* cf. *culcitae* MNHN-IA-2021-675 (5 mm in length, 1 mm maximum width without parapodia, 3 mm with parapodia). Photo credits: MNHN, G.F.

The animals were photographed alive and then preserved in ethanol 96%. DNA was extracted from tissue clips following a "salting-out" protocol [2] and the COI mitochondrial marker was amplified and sequenced for each specimen (following [15]). Two molecular datasets were assembled (for the worms and the snails) by retrieving similar COI sequences from GenBank (Supplementary Material Table S1). The datasets were used for calculating pairwise COI genetic distances and for producing Bayesian Inference phylogenetic trees (following [16]).

The two scale-worms formed a statistically supported clade with other specimens of *A. culcitae* collected in Japan (posterior probability = 1; Supplementary Material Figure S1), with COI genetic distances between the two groups (1.9–2.6%; Supplementary Material Table S2) compatible either with two geographically distant populations or with two distinct species. Scale-worm morphological characteristics were observed under a microscope (Zeiss Axio Stemi Lab A1) and compared with original descriptions of *A. culcitae* and *A. carlae* [13,14]. Specimen MNHN-IA-2021-676 measured 10 mm in length, with a maximum width of 2 mm without parapodia and 6 mm with parapodia; at least 18 segments and five pairs of elytra were countable (incomplete count). Specimen MNHN-IA-2021-675 measured 5 mm in length, with a maximum width of 1 mm without parapodia and 3 mm with parapodia; at least 22 segments and nine pairs of elytra were countable (incomplete count). The collected specimens showed features consistent with the diagnostic characteristics of *A. culcitae* (Figure 2). These included the morphology of the lower neurochaetae (Figure 2k,l) and elytra (Figure 2d), as well as the presence of micropapillae on elytra (Figure 2e). Considered both genetic and morphological information, the two scale-worm specimens were identified, for the purpose of this record, as *A.* cf. *culcitae*. However, further studies of additional specimens from the same geographic area and re-examinations of the *A. culcitae* type material, possibly including genetic sequences of the type material and additional molecular markers, are required to confidently state if these specimens represent a distant population of *A. culcitae* or a new species.

Based on Sugiyama et al.'s data [12] regarding color sexual dimorphism in *A. culcitae*, it could be hypothesized that specimen MNHN-IA-2021-676 (bigger and reddish) is a female, while MNHN-IA-2021-675 (smaller and whitish) is a male.

For the snail dataset, molecular data identified the collected specimen as a distinct lineage inside the genus *Coriocella* (Supplementary Material Figure S2) with a high genetic distance to the other species included in the dataset (13.7–19.7%; Supplementary Material Table S3). Morphological characteristics (dark mantle color and five dorsal warts) allowed its identification as *C.* cf. *tongana.*

This is the first time that *A.* cf. *culcitae* and *C.* cf. *tongana* have been found associated; it is also the first evidence of a symbiotic association between a lamellariin and an annelid, and the first record of a species of the genus *Asterophilia* ectosymbiotic on a mollusc host. Moreover, it extends the previously known distribution range of the genus *Asterophilia* down to New Caledonia.

Taking into consideration the limitation of this single, first record, it can still be hypothesized that *C.* cf. *tongana* may represent the one host on which *A. culcitae* forms mating pairs (along with asteroids). Moreover, at least in this observed case, the worms did not show a cryptic coloration with the host, but rather exhibited hiding behavior. The presence of secondary metabolites with strong cytotoxic activity in the mantle of *Coriocella* may represent one of the reasons why this genus is chosen as a host by scale-worms. The reported host range expansion regarding the genus *Asterophilia* parallels occurrences observed in other groups, such as the shrimps of the genus *Periclimenes* O.G. Costa, 1844 [17–20]. Therefore, unexplored associations may exist between polynoids and other velutinid species possessing defensive biocompounds, or with other gastropods exhibiting similar features, for example, some sea slug species. This record sheds light on the association between two groups of marine invertebrates, for which the ecology appears to be still largely unknown.

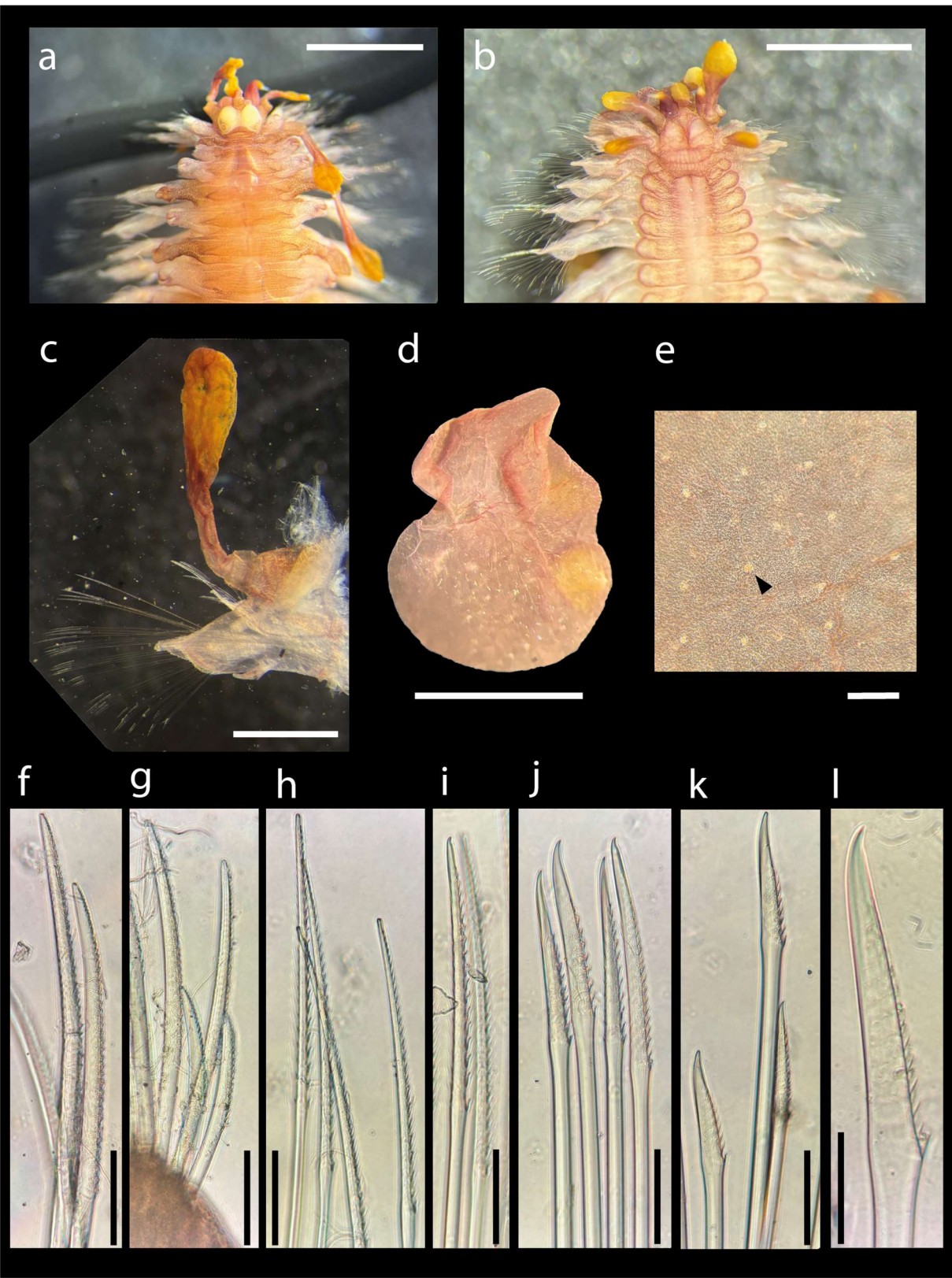

**Figure 2.** *Asterophilia* cf. *culcitae* MNHN-IA-2021-676. (**a**) Anterior end, dorsal view. (**b**) Anterior end, ventral view. (**c**) Parapodium of 8th chaetiger with dorsal cirrus, posterior view. (**d**) Elytron. (**e**) Detail of elytron with micropapillae (indicated by arrowhead). (**f,g**) Notochaetae. (**h**) Upper neurochaetae. (**i,j**) Middle neurochaetae. (**k,l**) Lower neurochaetae. Scale bars: (**a**–**d**) 1 mm; (**e**–**k**) 100 µm; (**l**), 25 µm. Photo credits: G.F.

**Supplementary Materials:** The following supporting information can be downloaded at: https://www.mdpi.com/article/10.3390/d16010065/s1, Table S1: Sequences used for molecular analyses; Table S2: Pairwise COI genetic distances between worm specimens; Table S3: Pairwise COI genetic distances between snail specimens; Figure S1: Bayesian Inference tree of the COI worm dataset; Figure S2: Bayesian Inference tree of the COI snail dataset.

**Funding:** This research was partially funded by Sapienza University of Rome SEED PNR 2021 grant n° B89J21032850001 to G.F.

**Institutional Review Board Statement:** Not applicable.

**Data Availability Statement:** COI sequences were deposited at GenBank (accession numbers: OR933634, OR933635, OR933636).

**Acknowledgments:** Specimens were collected under the auspices of the Our Planet Reviewed programme at the MNHN, during the Quinzaine des Nudibranches 2022 scientific expedition. The field work was made possible by funding from Fondation Gingko. It operated under Arrêté 3121-2022/ARR/DDDT issued by Valérie Gentien, Chef du Service de la Gestion des Aires Protégées, Province Sud. Special thanks go to Philippe Bouchet (MNHN) for the opportunity to work on New Caledonia material, to Catherine Le Bouteiller to have collected these specimens, to Barbara Buge (MNHN) for the precious help with MNHN vouchers, to Giacomo Chiappa (Sapienza University of Rome) for the help with molecular sequences, to Edoardo Casoli (Sapienza University of Rome) and Stephane Hourdez (CNRS-Sorbonne Université) for the help with worm identification, and to Romain Sabroux (University of Bristol) for the help with photographic material. Three anonymous reviewers were thanked for helpful suggestions.

**Conflicts of Interest:** The author declares no conflicts of interest.

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
