# Peer review of "Coriocella and the Worms: First Record of Scale-Worm Asterophilia cf. culcitae Ectosymbiotic on a Mollusc"

_diversity, doi:10.3390/d16010065_

Round 1
Reviewer 1 Report
Comments and Suggestions for Authors
This is an interesting short story about the unusual discovery of a sea star symbiont, the scaleworm Asterophilia culcitae, in symbiosis with the shellless caenogastropod Coricella cf. tongana. It is interesting that in this case the color pattern of the symbiont is not cryptic, as in conspecifics inhabited starfish. However, their safety is compensated by their cryptic behavior and the presence of metabolites with cytotoxic activity in the host’s mantle. This is the first report of a symbiotic association between lamellariin molluscs and polychaetes and the first report of a symbiosis between species of the genus Asterophilia and molluscs. The manuscript is interesting, well written and illustrated with high-quality photographs. I recommend publishing this manuscript in Diversity journal after minor revisions. My comments are given directly in the text of the manuscript.

Author Response
Answer: Thank you for the comments, bibliography references regarding the polynoid associations were added, typos were corrected, and it was explicitly added in the text that worm sex was only hypothesized based on colour and relative size.
Reviewer 2 Report
Comments and Suggestions for Authors
Dear Author,
Based on your results, it is impossible to make a clear conclusion that your specimens from New Caledonia belong to the species A. culcitae.
On the contrary, the results presented in Fig. 1 (Bayesian Inference tree of the worm dataset) suggest the existence of 2 closely related species of the same genus.
To draw a clear conclusion about the taxonomic status of your specimens collected in New Caledonia, further investigation of the sequences of other markers (16S, 18S, histone H3...) as well as the morphological characteristics of the worms is necessary.
Morphological part in this case is an integral and very important part of the work to achieve the stated goal within the framework of the stated article.
The method of molecular identification of species is currently only complementary and in no way can replace or cancel traditional morphological classification based on complex taxonomic analysis.
You cannot claim that your specimens belong to a particular species without clear evidence in the form of drawings/photographs of the species' morphological characteristics. The photographs of the general appearance of animals given in the article (Fig. 1), despite their colorfulness, do not provide any information about the characteristics that allow you to identify your specimens to a particular species.
Thus, in my opinion, the article as presented is not evidence-based and has no scientific value. Major revision is required to successfully publish your work.
Comments on the Quality of English LanguageI am not a native English speaker, but in my opinion the quality of English in this article is good.
Author Response
Thank you for your comments. The Byeasian analysis supported only one monophyletic clade including all the specimens (PP=1). The two sub-clades, even if present, were not statistically supported by this analysis (PP=0.83 for New Caledonia clade and 0.72 for Japan). However, the COI genetic distance around 2% could indicates two distinct species, but also two diversified populations (the other sequences available were all from Japan). Unfortunately, the only sequences of Asterophilia available in GenBank were of COI, so no comparison would have been possible, even sequencing additional markers for the new specimens. Moreover, 18S and H3 hardly accumulate mutation fast enough to be used to separate species. Worm identifications were integrated with morphological data following the original description of the two species of this genus. Diagnostic characters were photographed and added to the manuscript (Figure 2) and resulted compatible with A. culcitae (shape of the elytra, presence of micropapillae, shape of lower neurochaetae). Considered both COI genetic distance and morphological characters the identification was updated to A. cf. culcitae.
Reviewer 3 Report
Comments and Suggestions for Authors
The manuscript is a short message about the first find of ectosymbiosis in sea snail Coriocella and scale-worm Asterophilia. It is in fact interesting finding and I've no objections that research is dedicated only to own fact of finding. Nevertheless, I think, that it would be better not to have only a non-structured text, which in fact contains (in spite of its shortness) all traditional parts: introduction, materials and methods, results and discussion and conclusion.
Author Response
Thank you for your comments, regarding the overall structure of the short note I followed the guideline of “Interesting images from the sea” that specify to not dived the text in M&M etc, but I still tried to keep all the important information regarding each section.
Round 2
Reviewer 2 Report
Comments and Suggestions for Authors
Dear Author,
thank you for your work and for providing photographs of the worms as per previous recommendations.
The added photo raised new questions and recommendations regarding the correct identification of the worm species, which must be resolved for the successful publication of the article.
1 I was very surprised why the photo of the anterior end of the worm (fig. 2d) is presented only from the ventral side? Sorry, but any qualified taxonomist will ask you this question...... Agree that you will not be able to identify any animal only from the ventral side, for this you must at least look at it from the front (from dorsal in this case).
Of course, you can give a photo of the worm from the ventral side, which is also good, but only as an addition to the photo of the worm from the dorsal side!
Moreover, the prostomium should be clearly visible in the photo from the dorsal side; if it is covered by the first pair of elytra, then these elytra should be removed.
Parapodia are also an important taxonomic character, as are the prostomium, elytra and chaetae. Thus, in Fig. 2 should also present a cirrigerous parapodium with dorsal cirrus.
In its current version, Figure 2 is not acceptable for publication.
I recommend that you add a photo of the worm from the dorsal side, but leave the ventral view as well; a photo of a cirrigerous parapodium with dorsal cirrus; upper and middle neurochaetae; and notochaetae.
2 Please add information on pairs of elytra, number of segments, wide and long of worms, which can subsequently help researchers to correctly differentiate species if they turn out to be different or to correctly understand the scope of a species in the case of intraspecific variation in different populations.
3 Regarding the morphology of the lower neurochaeta shown in your figure 2c, I can see that it is similar to the neurochaeta depicted in figure 2O by Britaev and Fauchald, 2005.
But these authors stated that “Neurochaeta with an enlarged basal row of serration (resembling the semilunar pocket) was found in only one juvenile specimen” among the 29 specimens examined (1 holotype, 6 paratypes and 22 additional specimens; see Material examined in Britaev and Fauchald, 2005: 17). “Neurochaetae with an enlarged basal row of serration absent in most (obviously adult) specimens” (see Britaev and Fauchald, 2005: 20). Sugiyama et al., 2020 examining their 11 adult Japanese worms also found middle and lower neurochaetae with serrated edges that lacked semilunar pockets.
According to your results, you were dealing with adult worms. This is why it is important that you indicate the size of your worms and the number of segments.
I recommend that you check on your material how pronounced this feature is, i.e., how many such chaetae are there per parapodium and whether all parapodia have them along the body (there are many or few such parapodia....). It is very important to explore morphology of the upper and middle neurochaetae as well as notochaetae.
If this character is constant (there are many such chaetae and they are found in many parapodia) and if additional differences are found in other characters (upper - middle neurochaetae or notochaetae), then with a high degree of probability we can say that you have a new species. If there are not many such differences, but they are still obvious, you should note them and, on this basis, confirm your definition as cf.
In the latter case, it must be said that further study of both additional specimens from your area and re-examination of the type material is required, preferably with the support of molecular genetics based on not one but several markers.

Comments on the Quality of English LanguageI am not a native English speaker. In my opinion, the English is fine, but needs a little editing of the language.
Author Response
Thank you for the suggestions. Additional figures have been added, including the dorsal side of the worm, cirrigerous parapodium with dorsal cirrus, upper and middle neurochaetae, and notochaetae. In the text, details such as pairs of elytra, number of segments, width, and length of the worms have been included. The number of lower neurochaetae, similar to those in figure 2O by Britaev and Fauchald, 2005, were not abundant in the other parapodia checked. Therefore, I suggest retaining this preliminary identification as cf., especially considering it is based on two specimens only. As recommended, a paragraph has been added, stating that further investigation involving morphological and molecular data, including type material, is necessary to confidently determine whether this is A. culcitae or a new species. The manuscript has been revised by an English speaking colleague for polishing the form.
Round 3
Reviewer 2 Report
Comments and Suggestions for Authors
Dear author,
thank you for your detailed answers and the work done to improve the article. There are still some points in the article that require correction, which are outlined below.
Line 12-13 “However, so far only two instances have been reported thus far of crustaceans ectosymbiontic on their mantle.”
“so far” is the same as “thus far,” isn’t it? Please change the sentence, for example:
However, so far only two instances of crustaceans ectosymbiontic on their mantle have been reported.
Line 96-97 and 99-100 “After tissue clipping, at least 18 segments and five pairs of elytra were still countable (incomplete count).”; “After tissue clipping, at least 22 segments and nine pairs of elytra were still countable (incomplete count).”
Sorry, but how should I know how many segments you cut off for analysis? Moreover, the reader is not interested in how many segments you cut off, but how many of them the whole worm had....
The reader should not add up the number of cut off and remaining segments, he should simply read that the animal had so many body segments and elytra.
Please add relevant information here.
Line 181 “c: cirrigerous parapodium with dorsal cirrus.”
“cirrigerous” can be omitted here, the term itself means a bearing cirrus, I used it to emphasize your attention)).
Please, correct as: parapodium of chaetiger (add number of Ch) with dorsal cirrus, posterior view;
The segment number is necessary here and is always indicated, since the shape of the parapodia can change along the animal’s body (sometimes very strongly), parapodia of approximately identical segments need to be compared.
Line 182 “h, I, j: upper neurochaetae”
Unfortunately, In figs ‘h’ and ‘i’ I see upper neurochaetae with broken tips…..
Please find upper neurochaetae with whole tips and replace figs ‘h’ and ‘i’.
According to Britaev and Fauchald, 2005: “upper neurochaetae long, straight, slender, with 22-24 rows of serrations and notched, hooded tips”. It is very interesting and important to know which upper neurochaetes are present in your specimens.
As for fig. "j", I suppose it is a middle (not upper) neurochaeta with a bidentate tip and a shorter serrated part compared to the longer serrated part of upper neurochaetae. Which is consistent with Britaev’s description of the chaetae: “middle neurochaetae with relatively shorter serrated part and uni- or bidentate tips (Fig. 2K,L);”
Therefore, I suppose that figs ‘j, k, l’ as well as ‘m’ belong to middle neurochaetae, and only figs ‘n-o’ refer to lower neurochaetae.
Thus, 4 similar photos are for one type of neurochaetae…..
I suggest replacing the figs ‘h-I’ on figs of upper neurochaetae with whole tips; removing figs ‘l-m’; and leaving figs ‘j-k’ and ‘n-o’.
Comments on the Quality of English LanguageI'm not a native speaker, but I think the text still needs to be proofread.
Author Response
Line 12-13 “However, so far only two instances have been reported thus far of crustaceans ectosymbiontic on their mantle.”
“so far” is the same as “thus far,” isn’t it? Please change the sentence, for example:
However, so far only two instances of crustaceans ectosymbiontic on their mantle have been reported.
Answer: the sentence was corrected
Line 96-97 and 99-100 “After tissue clipping, at least 18 segments and five pairs of elytra were still countable (incomplete count).”; “After tissue clipping, at least 22 segments and nine pairs of elytra were still countable (incomplete count).”
Sorry, but how should I know how many segments you cut off for analysis? Moreover, the reader is not interested in how many segments you cut off, but how many of them the whole worm had....
The reader should not add up the number of cut off and remaining segments, he should simply read that the animal had so many body segments and elytra.
Please add relevant information here.
Answer: I am really sorry, but this is the only information I have (the incomplete count of segments and elytra). I edited the sentence to remove the unnecessary information.
Line 181 “c: cirrigerous parapodium with dorsal cirrus.”
“cirrigerous” can be omitted here, the term itself means a bearing cirrus, I used it to emphasize your attention)).
Please, correct as: parapodium of chaetiger (add number of Ch) with dorsal cirrus, posterior view;
The segment number is necessary here and is always indicated, since the shape of the parapodia can change along the animal’s body (sometimes very strongly), parapodia of approximately identical segments need to be compared.
Answer: corrected as suggested and added the info about the segment number (8th)
Line 182 “h, I, j: upper neurochaetae”
Unfortunately, In figs ‘h’ and ‘i’ I see upper neurochaetae with broken tips…..
Please find upper neurochaetae with whole tips and replace figs ‘h’ and ‘i’.
According to Britaev and Fauchald, 2005: “upper neurochaetae long, straight, slender, with 22-24 rows of serrations and notched, hooded tips”. It is very interesting and important to know which upper neurochaetes are present in your specimens.
As for fig. "j", I suppose it is a middle (not upper) neurochaeta with a bidentate tip and a shorter serrated part compared to the longer serrated part of upper neurochaetae. Which is consistent with Britaev’s description of the chaetae: “middle neurochaetae with relatively shorter serrated part and uni- or bidentate tips (Fig. 2K,L);”
Therefore, I suppose that figs ‘j, k, l’ as well as ‘m’ belong to middle neurochaetae, and only figs ‘n-o’ refer to lower neurochaetae.
Thus, 4 similar photos are for one type of neurochaetae…..
I suggest replacing the figs ‘h-I’ on figs of upper neurochaetae with whole tips; removing figs ‘l-m’; and leaving figs ‘j-k’ and ‘n-o’.
Answer: figure 2 was updated following suggestions; neurochaetae photos were replaced, and figure l-m removed.